# Impact of Incorporating Defatted Black Soldier Fly Meal into Diet on Growth Performance, Serum Biochemical Parameters, Nutrient Digestibility, Morphology of the Intestinal Tract, and Immune Index of Brooding Laying Hens

**DOI:** 10.3390/ani15050625

**Published:** 2025-02-20

**Authors:** Lusheng Li, Lifei Chen, Guiying Wang, Yinling Zhao, Yizhen Xin, Meng Xu, Yuxi Wang, Hanhan Song, Jiani Fu, Rongsheng Shang, Jibin Zhang

**Affiliations:** 1College of Life Science and Technology, Huazhong Agricultural University, Wuhan 430070, China; srs19990728@163.com; 2College of Agriculture and Biology, Shandong Province Engineering Research Center of Black Soldier Fly Breeding and Organic Waste Conversion, Liaocheng University, Liaocheng 252000, China; chenlifei@lcu.edu.cn (L.C.); wangguiying@lcu.edu.cn (G.W.); 2210190117@stu.lcu.edu.cn (Y.X.); 2210190104@stu.lcu.edu.cn (M.X.); ozhitiano@outlook.com (Y.W.); 17852761158@163.com (H.S.); 15263633726@163.com (J.F.); 3Shandong Fengxiang Co., Ltd., Liaocheng 252323, China; hainuo1825@163.com

**Keywords:** black soldier fly larvae meal, brooding laying hens, growth performance, serum biochemical indices, antioxidant capacity

## Abstract

This study explored the use of defatted black soldier fly larvae meal (BSFM) as a sustainable protein alternative to soybean meal in the diets of brooding laying hens. Over 42 days, 480 chicks were divided into four groups fed diets with 0%, 3%, 6%, or 9% BSFM. Results showed that birds fed 3% BSFM had higher daily weight gain and improved feed efficiency during early growth phases. BSFM supplementation also enhanced blood protein levels, antioxidant capacity, and immune function while reducing fat and nitrogen waste in the blood. Nutrient digestion and gut health remained stable, with no negative impacts on feed intake or bone development. These findings suggest that BSFM can effectively replace soybean meal in poultry feed, offering a sustainable protein source that supports animal growth, health, and environmental goals. This research provides valuable insights for reducing reliance on traditional protein sources in the poultry industry.

## 1. Introduction

The continuous growth of the global population and higher demands from consumers to increase sustainable animal production have elevated the need for protein ingredients [1]. Plant protein shortages have become increasingly prominent worldwide [2]. As of 2023, the People’s Republic of China’s total imports of soybeans amounted to 994,090 thousand tons, which constituted in excess of 84% of the domestic supply during the corresponding period [3]. The demand for alternative protein sources for animal husbandry is becoming increasingly urgent. Thus, exploring alternative protein sources is a viable solution for reducing the dependence on traditional protein sources.

Such an innovative alternative protein source is edible insects, which are characterized by a large biomass, high reproduction and biological conversion efficiency, wide feeding range, and require less land space [4,5]. Moreover, some insects can be reared on various types of organic waste streams [6,7,8]. The protein levels and essential amino acid profile of insects meet or are well above the FAO/WHO dietary requirements, and they also meet the nutritional requirements for the growth of livestock and poultry [5,9]. The consumption of edible insects has been posited as a potential avenue for augmenting the protein content of one’s diet, on account of the higher quality of protein they are reputed to contain as compared to more conventional dietary sources [10].

The black soldier fly (BSF), scientifically known as *Hermetia illucens* L., is counted among the insect species acknowledged as some of the most promising alternative protein sources for animal feed production [11]. In recent years, it has garnered significant attention for its potential within the framework of circular organic waste management. This is attributed to its rapid conversion capabilities and the high nutritional value of its prepupa. The larvae of the BSF play a crucial role as feed in animal production, thereby highlighting their importance in sustainable waste management [12,13]. These larvae contain high levels of proteins, accounting for 40–45% of the dry matter, and fat, making up 25–40% of the dry matter. As a result, they can be effectively utilized as feedstuff for various livestock, including poultry, swine, and fish. Studies have shown that the application of BSF-related products has achieved good results in broilers [14], Muscovy ducks [15], layers [16], commercial pigs [17], and other livestock and poultry.

As a new type of protein feed, BSF meal (BSFM) is mostly used to replace soybean or fish meal in equal proportions in the feed of laying hens [18,19,20]. Part of the study was carried out using the energy conversion method, but the energy data used was total energy rather than common metabolic energy. A small amount of live insects was added directly [21,22]. In terms of the usage effect, there are also great differences in the effects of different methods of use. Some studies have shown that replacing 15% soybean meal with BSFM has no adverse impact on the productivity of hens [23]. Some reports have also shown that BSFM does not reduce the performance of laying hens when completely replacing soybean meal [24,25]. However, Khan et al. [26] found that though the addition of 9% BSFM did not affect the production performance of laying hens, the addition of 18% BSFM significantly reduced their laying performance, indicating that excessive BSFM would affect the laying performance of laying hens. Most studies have reported consistent results. These findings might be attributed to the lack of clarity about nutrient compositions.

Currently, there is limited literature on the metabolizable energy and nutrient digestibility data of BSFM, with most studies focusing on broilers. Mahmoud et al. determined the metabolizable energy value of full-fat BSFM for broilers using an alternative method, reporting a value of 19.10 MJ/kg [27]. De Marco et al. [28] found a metabolizable energy of 17.38 MJ/kg with a similar diet. Schiavone et al. [29] determined the metabolizable energy of partially defatted BSFM to be 16.25 MJ/kg. In another study, growing laying hens were selected as the research animals, and the metabolizable energies of BSFM with varying degrees of defatting were determined to be 16.34 MJ/kg and 12.41 MJ/kg, respectively. The results indicated a strong positive correlation between metabolizable energy and the degree of defatting [30]. These findings indicated significantly variable data on the metabolizable energy of BSFM, closely associated with the test animals, determination methods, and BSFM characteristics. However, no relevant reports have been published regarding feeding trials utilizing this data. The brooding period of laying hens ranges from birth to 6 weeks of age. Early feeding has been shown to positively affect the growth, development, and maturation of the gastrointestinal tract and egg-laying performance of the hens [31]. Protein quantity and quality are positively and significantly associated with body weight and tibia length during the brooding period of laying hens, which are closely related to their laying performance [32]. Therefore, it is essential for laying hens to find alternative protein sources. To the best of our knowledge, fewer studies have yet to explore the potential of BSF larvae as a replacement for fish/soybean meal during the brooding period of laying hens. The present study was aimed at assessing the effects of BSFM inclusion in the feed of brooding laying chickens in terms of their production performance, blood biochemical indices, antioxidant capacity, nutrient digestibility, and immune function.

## 2. Materials and Methods

### 2.1. Preparation of BSF Larva Powder

Five-day-old BSF larvae were procured from Shandong Woneng, Agricultural Technology LLC (Liaocheng, China). Household organic waste was utilized as a source of sustenance for BSF larvae. The larvae were introduced into the experimental household organic waste at a seeding rate of 1 larva per gram. Subsequently, they were reared on this feed for a duration of 10 days, with the temperature maintained within the range of 25–30 °C. The live larvae were separated and dried to 5% in an oven at 80 °C after cleaning. After crushing, they were wrapped with filter paper and placed into an extraction tank. After extracting the insect meal for 5 h, as previously described by Kong et al. [33], it was crushed again to prepare degreased BSFM. The nutrient composition and metabolizable energy of the BSFM were obtained from Xin et al. [30].

### 2.2. Animals and Experimental Design

A total of 480 one-day-old chicks (Hy-Line Brown, commercial flock) were randomly allotted to four groups based on their dietary treatments, with each comprising six pens as replicates with 20 chicks/pen. The chicks were reared until the end of the brooding period set at 42 days. The basal diet was developed in line with the nutritional needs of Hy-Line Brown laying hens during the brooding stage. The first group received only the basal diet, which was designated as G0. For the other three groups, their basal diets were supplemented with 3%, 6%, or 9% BSFM. These groups were respectively labeled as the G3, G6, and G9 treatment groups. All diets were isonitrogenous and isocaloric to meet the brooding period nutrient requirements and were adjusted according to the BSFM nutritional composition (Table 1). The diets were fed in pellet form.

### 2.3. Bird Husbandry

The present study was conducted in accordance with the methods approved by the Animal Care and Use Committee of Liaocheng University, China (AP2024022927). The experiments were carried out in the agricultural ecological park of Liaocheng, University. Prior to the commencement of the experiments, the rearing house and equipment were subjected to a thorough fumigation process using a standard concentration of formaldehyde and potassium permanganate solution. This procedure was conducted three days prior to the initiation of the experiments. When the chicks arrived, each one was tagged on the wing, had its weight measured, and was randomly allocated to one of the four groups. During the initial three days, the temperature was maintained at 33 °C. After that, the temperature was steadily decreased at a rate of 3 °C per week. By the time the chicks reached 21 days old, the temperature had been reduced to 21 °C. An enclosed chicken coop was adopted, with each replicate using one chicken cage (length × width × height: 87 × 45 × 24 cm) and artificial ventilation was employed. In the first three days, 24-h lighting was provided, and afterward, the lighting duration was maintained at 10 h per day. White light tubes were used for illumination. Feed and water were available to the chicks at all times during the entire trial period. During the experiments, routine immunizations and medications were administered to the birds, the daily feed amount was recorded, and the health status of the chicks was monitored.

### 2.4. Assessment of Growth Performance

At 42 days of age, prior to the conclusion of the experiment, the chicks were subjected to a 12-h fast. The body weight of each replicate was measured and documented. These measurements were then used to calculate the average daily gain (ADG), average daily feed intake (ADFI), and feed conversion ratio (FCR) of the birds. Additionally, the birds’ weights were recorded on the first day of the experiment. The birds were provided with feed on a daily basis, and the amount of feed remaining was meticulously recorded to calculate feed efficiency.

### 2.5. Determination of Serum Biochemical Parameters

At 21 and 42 days of age, two birds were randomly chosen from each replicate to analyze their serum biochemistry. The birds were fasted for 12 h, after which 4 mL of blood was collected from the wing vein of each bird. The blood samples were then centrifuged at 4 °C at 4000 rpm for 15 min to obtain plasma. The separated sera were stored in a −80 °C freezer until further analysis. For the 42-day-old birds, an automated clinical chemistry analyzer (Cobas-8000 c702, Basel, Switzerland) was employed to measure various serum biochemical parameters. These included aspartate transaminase (ALT) levels, alanine aminotransferase (AST) levels, total protein, albumin, and globulin levels, as well as blood levels of urea nitrogen, uric acid, glucose, triglycerides, and cholesterol. To determine the total antioxidant capacity (T-AOC), glutathione peroxidase (GSH-Px) activity, malondialdehyde (MDA) levels, and total superoxide dismutase (T-SOD) activity in 21- and 42-day-old birds, commercial kits from the Nanjing Jiancheng Institute of Bioengineering (Nanjing, China) were used. All measurements were carried out following the manufacturer’s instructions.

### 2.6. Determination of Nutrient Digestibility

Nutrient digestibility was analyzed using the total fecal collection method. Briefly, after 42 days, two healthy chickens with similar weights were randomly selected from each replicate and reared individually in clean and disinfected metabolic cages. The animals were provided with food and water ad libitum. The experimental period lasted for a period of 5 days. Chicks in each group were fed their respective diets, while feed consumption was accurately recorded daily during the whole experimental period. Their droppings were collected for four days, which was deemed the collection period. The collection was performed at approximately 8:00 a.m. each morning prior to the provision of the next daily ration. For each replicate, feces collected during the 4 days were mixed thoroughly and dried to a constant weight at 65 °C after fixation of nitrogen with 10% hydrochloric acid. Each fecal mixture was ground to 40 mesh and stored at −20 °C until further use.

Crude protein content, crude fat content, calcium levels, and total phosphorus levels in the feces were determined using the Kjeldahl (GB/T 6432-2018) [34], Soxhlet extraction (GB/T 6433-2006) [35], potassium permanganate (GB/T 6436-2018) [36], and colorimetric methods (GB/T 6437-2018) [37], respectively.

### 2.7. Intestinal Mucosal Immune Index

Following the nutrient digestibility experiment, one bird from each replicate was sacrificed via cervical dislocation. The middle and lower parts of the ileum were washed with a saline solution to eliminate its contents. Using the edge of a glass slide, one gram of the intestinal mucosa was carefully scraped and stored at −80 °C pending further analysis. To measure the levels of secreted immunoglobulin A (sIgA), interleukin-2 (IL-2), IL-6, and tumor necrosis factor (TNF)-α, ELISA kits from the Nanjing Jiancheng Institute of Bioengineering (Nanjing, China) were utilized. All measurements were performed in strict accordance with the manufacturer’s instructions.

### 2.8. Statistical Analysis

For the data collected during this study, initial sorting was carried out with Excel 2019. Then, to conduct a more in-depth analysis, one-way ANOVA and Tukey’s multiple comparison of variance were performed using SPSS software (version 22.0, SPSS for Windows, release 19.0, developed by SPSS Inc., Chicago, IL, USA). In order to distinguish statistically significant differences among the means, orthogonal polynomials were used to test linear, quadratic, and cubic responses to dietary levels of BSFM. Finally, the data are presented in the format of mean ± standard deviation (SD), and statistical significance is reported at *p* < 0.05.

## 3. Results

### 3.1. Effects of BSFM on the Growth Performance of Chicks

As presented in Table 2, during the 1–21-day, 22–24-day, and 1–42-day phases, the ADG of the G3 group was notably higher than that of the G0 group (*p* < 0.05). Meanwhile, the ADG of the G6 group was also higher than that of the G0 group, yet the difference was only marginal (*p* > 0.05). Regarding the FCR, during the 1–21-day phase, the FCR of the G3 group was significantly lower compared to the G0 group (*p* < 0.05). However, in other phases, there were no significant differences in FCR among the other groups (*p* > 0.05). Furthermore, when examining ADFI and tibial lengths across all groups, no significant differences were detected (*p* > 0.05).

### 3.2. Effects of BSF on Serum Biochemical Parameters of Chicks

The serum biochemical parameters of the chicks are shown in Table 3. The serum biochemical parameters differed significantly across the groups. Compared with the G0 group, the total protein and globulin levels were significantly higher, while the triglyceride levels were significantly lower in the G3 and G6 groups (*p* < 0.05). In addition, the urea nitrogen levels were significantly lower in the G3 group than in the G0 group (*p* < 0.05). AST and ALT activities, as well as the levels of uric acid, cholesterol, and glucose, were lower in the treated groups than in the G0 group, but only mildly (*p* > 0.05).

### 3.3. Effects of BSF on the Antioxidant Capacity of Birds

As indicated in Table 4, at both 21 and 42 days of age, the G6 and G9 groups demonstrated significantly higher T-AOC levels compared to the G0 group (*p* < 0.05). Although the G3 group also had higher T-AOC values than the G0 group, the difference was not significant (*p* > 0.05). Moreover, all treated groups showed significantly lower MDA levels in comparison to the G0 group (*p* < 0.05). Regarding the GSH-Px activities of 21-day-old birds, the G3, G6, and G9 groups had significantly higher values than the G0 group (*p* < 0.05). Finally, upon examination, the T-SOD activities were found to be similar across all groups (*p* > 0.05).

### 3.4. Effect of BSFM on Nutrient Digestibility

As shown in Figure 1, the apparent digestibility of crude fat in the G3 group and the G6 group was significantly higher than that in the G0 group (*p* < 0.05). There was no significant difference in apparent digestibility of crude protein, calcium and phosphorus among groups (*p* > 0.05).

### 3.5. Effects of BSF on the Intestinal Mucosal Immunity of Chicks

As depicted in Figure 2, the treated groups had significantly higher secreted sIgA levels compared to the G0 group (*p* < 0.05). Nevertheless, there were no significant differences in sIgA levels among the treated groups themselves. All treated groups showed higher IL-2 levels than the G0 group, yet these differences were not statistically significant (*p* > 0.05). Finally, upon assessment, the G9 group has a significant difference designation in the IL-6 levels compared with the G0 group (*p* < 0.01); the IL-6 and TNF-α levels were found to be comparable across the other groups (*p* > 0.05).

## 4. Discussion

In the present study, we investigated the effects of the inclusion of BSFM powder in the diet of laying hens during the brooding period. Our results showed that the inclusion of 3% BSFM in the feed improved the ADG of the laying hens throughout the brooding period, and FCR decreased during the 1–21 day phase. However, BSFM did not affect the ADFI and tibial lengths of the laying hens. These findings indicated that the diet supplemented with 3% BSFM did not influence the growth performance but could be used as a suitable alternative protein source for brooding laying hens. Dorper et al. [14] used live BSF larvae and full-fat and partially defatted BSFM products instead of soybean meal to feed broilers and reported improved growth performance of the chickens without affecting carcass quality. Dabbou et al. [38] analyzed the effects of replacing soybean meal with BSFM on broiler chickens. They reported that the inclusion of 5% and 10% BSFM in the feed did not affect the ADG and FCR of the birds; however, the inclusion of 15% BSFM in the feed decreased and increased their ADG and FCR, respectively. Chu et al. [12] also reported that the inclusion of 3% and 6% BSFM in the diet did not affect the ADG and FCR of chickens. The present study also reported no significant effects of the inclusion of 3% BSFM in the feed on the ADG and FCR of the chickens. However, our findings were inconsistent with the findings of Dabbou et al. [38]. The discrepancy might be attributed to the differences in the nutritional composition of the BSFM. Schiavone et al. [29] assessed the effects of defatted BSFM on the nutrient digestibility of broilers and reported that the nutrient digestibility of the group fed with low levels of defatted BSFM was significantly higher than that of the group fed with high levels of defatted BSFM. The nutritional value of BSFM varies greatly depending on its source [33,38], and different sources of BSFM might lead to varying growth performances of animals. Numerous studies have shown that an appropriate amount of BSFM can improve the growth performance of animals and successfully replace (at least partly) fish meal, soybean meal, and other raw protein sources [39,40]. However, increasing the inclusion of BSFM in the feed to a certain degree decreases the growth performance of animals [11]. Our results were in agreement with the findings of the above-mentioned studies. Similar results, with minor discrepancies, have been reported with the application of house fly, yellow mealworm, and other insect feeds [41,42]. The decrease in the growth performance of animals with increasing BSFM proportion might be attributed to the protein levels of the basic formula, the types of animals, nutritional composition, apparent digestibility, amino acid composition, and palatability [43,44].

Serum biochemistry reflects the physiological state, metabolic condition, and cell permeability of animals, which are important indicators for assessing their physical condition [45]. Total serum protein comprises albumin and globulin. In birds, similar to mammals, a total serum protein test is used to evaluate the nutritional status and hepatic function. Globulins are produced by the liver and the immune system. Elevated globulin levels are likely associated with the maturation of the immune system, and high total serum protein levels can stimulate animal protein synthesis and growth [46]. Blood urea nitrogen is a serum byproduct of protein metabolism. Its concentration can be used as an indicator of nitrogen balance and protein metabolism. Low blood urea nitrogen levels indicate that the animal is highly efficient in using amino acids for protein accretion [47,48]. In contrast, high blood urea nitrogen levels are associated with decreased protein deposition and a low protein utilization rate.

Gariglio et al. [49] reported that 6% defatted BSFM significantly increased the total serum protein level of Muscovy ducks. Dabbou et al. [38] showed that 10% defatted BSFM reduced the serum uric acid levels of broilers. In the current study, the 3% and 6% BSFM significantly increased the total serum protein and globulin levels and significantly reduced blood urea nitrogen levels. These results indicated that BSFM can improve the protein utilization rate of the laying hens during the brooding stage, enhancing their immune function and growth performance. The findings of this study aligned with those of previous research.

Serum triglyceride and cholesterol levels reflect the absorption and metabolism of lipids. In the present study, BSFM administration did not influence glucose levels. However, the serum triglyceride levels decreased. A previous study on growing pigs reported that 4% and 8% BSFM significantly elevated blood glucose levels but did not alter cholesterol or triglyceride levels [12]. Our findings indicated that BSFM supplementation enhanced the body’s metabolism, increased protein deposition, improved lipid metabolism, and promoted the growth of laying hens.

The animal body is equipped with an antioxidant system that counteracts the damaging effects of oxidative free radicals. The antioxidant system consists of antioxidant enzymes and non-enzymatic antioxidants. T-AOC is a comprehensive index that evaluates the total antioxidant status of the animal body and the overall response of non-enzymatic antioxidants and antioxidant enzymes. T-SOD, MDA, and GSH-Px are the most widely reported indices directly or indirectly reflecting the functional status of the antioxidant system [50]. Consistent with the findings of previous studies, we also observed elevated T-AOC and decreased GSH-Px activities in BSFM-fed groups. Dabbou et al. [38] found that defatted BSFM significantly improved GSH-Px activity in broilers in a dose-dependent manner. Gariglio et al. [49] reported that Muscovy ducks fed 9% defatted BSFM exhibited significantly lower MDA levels but comparable GSH-Px activity compared to the control group. Li et al. [51] reported that defatted BSFM had no significant effect on T-SOD and MDA activities, but 75% or 100% BSFM in the diet significantly increased serum catalase activity in Jian carp. These results showed that BSFM might boost the antioxidant capacity of animals. Secci et al. [52] reported that BSFM is rich in dietary tocopherols (42.72 mg/kg of insect meal), which might promote GSH-Px activity. Based on these findings, we hypothesized that the elevated antioxidant capacity of BSFM-fed chickens might be associated with the inclusion of tocopherols in the insect meal. Further research is needed to better understand the relationship between the biologically active components of BSFM and the antioxidant capacity of animals.

Furthermore, our findings showed that the inclusion of BSFM in the diet of brooding laying hens did not influence the apparent digestibility of dry matter, organic matter, calcium, and phosphorus. These findings were also in line with the results of previous studies. Gariglio et al. [49] reported that the digestibility of dry matter and organic matter in Muscovy ducks was not affected by BSF larva meal inclusion levels. Shang et al. [11] reported no differences in the apparent digestibility of calcium and magnesium in growing pigs based on varying BSFM inclusion levels in the diet. Similar results were observed in broiler quail when soybean meal was partially replaced with BSF larva meal [53]. On the contrary, Yu Miao et al. showed that the inclusion of 4% BSFM in the diet of growing pigs significantly improved the digestibility of organic matter. In contrast, Cutrignelli et al. [54] observed that the inclusion of 17% BSFM in the diet of laying hens reduced the apparent digestibility of dry matter. Bovera et al. [55] showed that the inclusion of 7.3% BSFM in the diet of laying hens did not affect the apparent dry matter digestibility; however, 14.6% BSFM reduced apparent dry matter digestibility.

Our results showed that 3% and 6% BSFM significantly improved the apparent digestibility of crude protein and crude fat in laying hens. However, protein and fat digestibility tended to decrease in the group fed 9% BSFM, which was consistent with their significantly decreased growth performance.

Gariglio et al. [49] showed that protein digestibility and fat digestibility improved with increasing BSFM proportions in animal feed. Some studies have shown that BSFM reduced the protein digestibility of laying hens but did not affect their fat digestibility [38,52]. Cullere et al. found that BSFM significantly improved the apparent fat digestibility of broiler quail [53].

BSFM is known to contain a substance called chitin, which has a negative effect on protein and lipid digestion, especially in animals that lack chitinolytic activity. Chitinase production in the proventriculus and hepatocytes has been demonstrated in chickens [56]. However, their chitinase levels are dependent on their feeding behaviors and feed ingredients, impacting their chitin digestibility [57]. Our findings demonstrated that protein digestibility in chickens decreased with increasing BSFM proportions in the feed.

The fatty acid profile reflects the quality of the utilized lipid and the apparent fat digestibility of birds [58]. Generally, BSF larvae are rich in palmitic acid, oleic acid, and lauric acid (11–16%, 12–32%, and 20–40% of total lipids, respectively), which is very similar to the composition of coconut oil fatty acids [59]. Schiavone et al. [29] found that the fat digestibility of the group fed low levels of defatted BSFM was significantly higher than that of the group fed with high levels of defatted BSFM. Gariglio et al. [49] found that fatty acid digestibility in Muscovy ducks decreased with increasing BSFM levels in their diet. Thus, we speculated that the levels of chitin and saturated fatty acids influence the digestion and absorption of BSFM in poultry; however, the underlying mechanism is still unclear and needs further exploration.

The intestinal tract represents the largest immune organ in animals, functioning as a barrier to prevent the passage of harmful intraluminal entities, including foreign antigens, microorganisms, and their toxins. The most abundant adaptive immune factor in the intestinal lumen is sIgA, which is produced by plasma cells and epithelial cells overlying mucosal surfaces. sIgA fulfills a pivotal function as the primary line of defense by neutralizing toxins and viruses present in the mucosa and preventing the adhesion of pathogens to mucosal epithelial cells [60,61]. Pro-inflammatory cytokines such as IL-2, IL-6, and TNF-α play crucial roles in intestinal mucosal immunity. In the current study, the titers of sIgA in the groups fed BSFM were observed to be significantly higher than those in the control group. Additionally, the mean levels of IL-2 in the treated groups were higher than those in the control group, although this difference did not reach statistical significance (*p* > 0.05). Moreover, the levels of IL-6 and TNF-α were comparable between the treated and control groups.

As far as we are aware, no previous studies have investigated the impact of BSFM on the immune indices of brooding laying hens as described above. This research thus contributes to filling a gap in understanding the effects of alternative feed sources like BSFM on the immune function of laying hens during the brooding stage. However, several studies have reported that an appropriate amount of BSFM might improve the antioxidant capacity and gut mucosal structure of poultry. BSFM is most commonly speculated to mediate its effects via a mechanism associated with the content levels of chitin, lauric acid, and antimicrobial peptides [62].

## 5. Conclusions

The diet supplemented with 3% defatted BSFM significantly increased the ADG, serum total protein and globulin levels and crude protein digestibility of the chicks but decreased the FCR of chicks aged 1–21 days. Additionally, the inclusion of defatted BSFM in the diet led to a linear increase in GSH-Px and T-SOD activities, as well as ileum mucosal sIgA levels, in chickens aged 42 days, thereby enhancing their antioxidant capacity and ileum mucosal immunity. The positive outcomes in terms of enhanced growth performance and gastrointestinal health indicate that incorporating defatted BSFM as a protein source in the diets of young laying chickens is a viable and better feed alternative.

## Figures and Tables

**Figure 1 animals-15-00625-f001:**
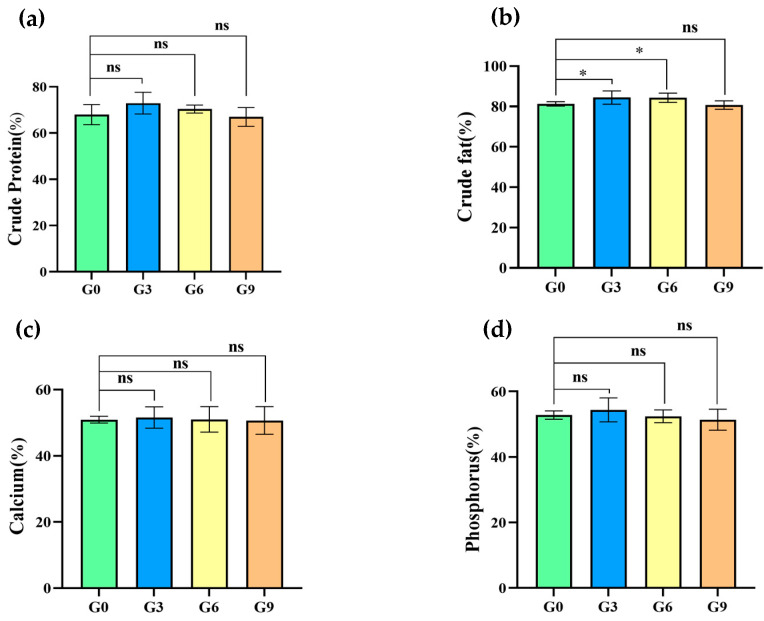
Effects of BSF on apparent digestibility. (**a**) Crude protein; (**b**) crude fat; (**c**) calcium; (**d**) phosphorus. The symbol “ns” indicates that the difference is not significant (*p* > 0.05); the symbol “*” represents that the difference is significant (*p* < 0.05).

**Figure 2 animals-15-00625-f002:**
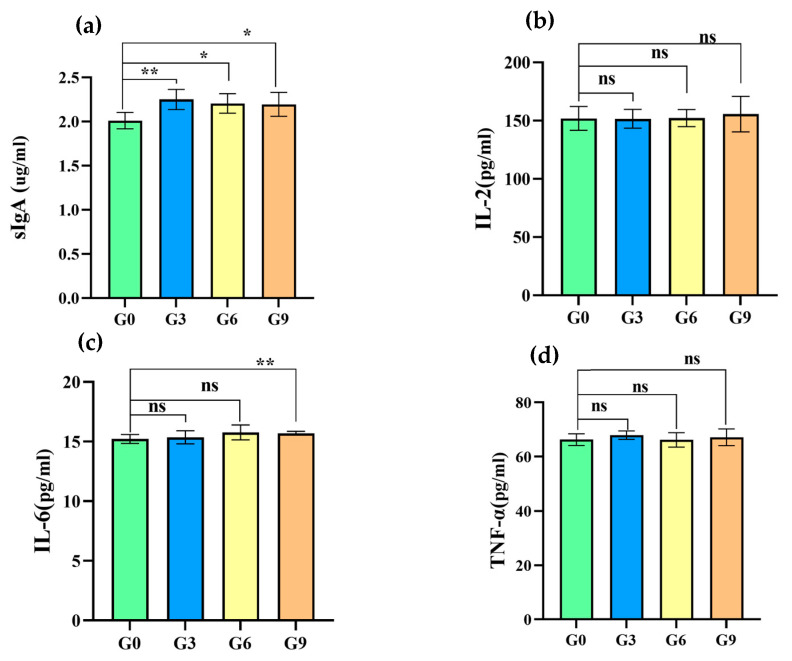
Effects of BSF on intestinal mucosal immunity. (**a**) sIgA; (**b**) IL-2; (**c**) IL-6; (**d**) TNF-α. The symbol “ns” indicates that the difference is not significant (*p* > 0.05); the symbol “*” represents that the difference is significant (*p* < 0.05); and the symbol “**” indicates that the difference is extremely significant (*p* < 0.01).

**Table 1 animals-15-00625-t001:** Composition and nutrient levels of experimental diets (as-fed basis).

Items (%)	Inclusion Levels ^1^, %			
	0	3	6	9
Ingredients				
Corn	65.01	67.66	67.55	67.21
Soybean meal	26.37	22.30	22.63	20.07
Corn gluten meal	3.00	3.00	0.44	0.00
Defatted BSFM	0.00	3.00	6.00	9.00
Soybean oil	1.35	0.00	0.00	0.00
Limestone	1.40	1.09	0.54	0.83
Dicalcium phosphate	1.50	1.58	1.64	1.76
L-Lysine·HCl	0.27	0.28	0.10	0.04
DL-Methionine	0.10	0.09	0.10	0.09
Vitamin and mineral premix	1.00	1.00	1.00	1.00
Total	100.0	100.0	100.0	100.0
Nutrient levels *				
Metabolizable energy, MJ/kg	12.13	12.13	12.13	12.13
Crude protein, %	19.00	19.00	19.00	19.00
Calcium, %	0.87	0.98	1.00	1.30
Non-phytate phosphorus, %	0.41	0.42	0.42	0.43
Lysine, %	1.05	1.05	1.05	1.05
Methionine, %	0.41	0.41	0.41	0.41

* The crude protein, lysine, and methionine levels were measured values; other nutrient levels were calculated values. BSFM = black soldier fly larvae meal. ^1^ G0, diet supplemented with 0% BSFM; G3, diet supplemented with 3% BSFM; G6, diet supplemented with 6% BSFM; G9, diet supplemented with 9% BSFM.

**Table 2 animals-15-00625-t002:** Effects of different black soldier fly larvae meal levels on the growth performance of chickens.

Items ^1^	Groups ^2^			
	G0	G3	G6	G9
1 to 21 days of age				
ADG/(g/d)	6.20 ± 0.12 ^a^	6.68 ± 0.12 ^b^	6.59 ± 0.15 ^ab^	6.21 ± 0.10 ^a^
ADFI/(g/d)	18.92 ± 0.22	18.95 ± 0.41	19.01 ± 0.22	19.01 ± 0.19
FCR	3.05 ± 0.01 ^a^	2.84 ± 0.02 ^b^	2.88 ± 0.01 ^ab^	3.06 ± 0.02 ^a^
TL/(mm)	45.81 ± 0.38	45.90 ± 0.52	45.96 ± 0.62	45.89 ± 0.84
22 to 42 days of age				
ADG/(g/d)	14.17 ± 0.06 ^a^	14.93 ± 0.09 ^b^	14.48 ± 0.10 ^ab^	14.49 ± 0.09 ^ab^
ADFI/(g/d)	26.56 ± 0.36	27.05 ± 0.41	26.29 ± 0.48	26.34 ± 0.45
FCR	1.87 ± 0.03	1.81 ± 0.02	1.82 ± 0.04	1.82 ± 0.02
TL/(mm)	23.43 ± 1.99	24.49 ± 1.47	24.24 ± 1.30	24.13 ± 1.16
1 to 42 days of age				
ADG/(g/d)	10.19 ± 0.15 ^a^	10.81 ± 0.11 ^b^	10.54 ± 0.10 ^ab^	10.35 ± 0.09 ^ab^
ADFI/(g/d)	22.74 ± 0.46	23.00 ± 0.27	22.65 ± 0.49	22.68 ± 0.42
FCR	2.23 ± 0.03	2.13 ± 0.02	2.15 ± 0.03	2.19 ± 0.02
TL/(mm)	69.24 ± 1.23	70.39 ± 1.05	70.20 ± 1.00	70.02 ± 1.12
SR/(%)	92.85 ± 1.31	93.13 ± 1.58	94.14 ± 2.07	93.52 ± 1.63

^a,b^ Means that the values within a row with no common letters differ significantly (*p* < 0.05). ^1^ ADFI = average daily feed intake; ADG = average daily gain; FCR = feed conversion ratio; SR = Survival rate; TL = Toe length. ^2^ G0, diet supplemented with 0% BSFM; G3, diet supplemented with 3% BSFM; G6, diet supplemented with 6% BSFM; G9, diet supplemented with 9% BSFM. Values are mean ± SEM (*n* = 6).

**Table 3 animals-15-00625-t003:** Effects of different black soldier fly larvae meal levels on the serum biochemical indices of chickens.

Items ^1^	Groups ^2^			
	G0	G3	G6	G9
TP/(g/L)	41.12 ± 2.01 ^a^	49.34 ± 1.41 ^b^	51.87 ± 2.40 ^b^	43.85 ± 2.84 ^a^
ALB/(g/L)	16.01 ± 0.33	18.92 ± 0.57	18.89 ± 1.33	16.51 ± 1.04
GLO/(g/L)	25.10 ± 1.08 ^a^	29.58 ± 0.80 ^b^	30.45 ± 2.11 ^b^	27.31 ± 1.45 ^ab^
UN/(mmol/L)	0.36 ± 0.01 ^a^	0.29 ± 0.01 ^b^	0.30 ± 0.01 ^ab^	0.33 ± 0.01 ^a^
UA/(μmol/L)	181.55 ± 13.61	162.58 ± 20.79	169.28 ± 25.61	175.20 ± 24.91
TC/(mmol/L)	4.04 ± 0.20	3.62 ± 0.25	3.65 ± 0.27	3.80 ± 0.24
TG/(mmol/L)	0.47 ± 0.14 ^a^	0.32 ± 0.03 ^b^	0.34 ± 0.02 ^b^	0.45 ± 0.21 ^a^
GLU/(mmol/L)	13.66 ± 0.37	13.71 ± 0.30	13.58 ± 0.32	13.60 ± 0.12
ALT/(U/L)	26.83 ± 1.22	24.95 ± 1.50	25.24 ± 0.25	26.63 ± 2.47
AST/(U/L)	181.72 ± 10.21	175.34 ± 5.04	168.23 ± 5.78	175.34 ± 14.21

^a,b^ Means that the values within a row with no common letters differ significantly (*p* < 0.05). ^1^ TP = total serum protein; ALB = albumin; GLOB = globulin; CHOL = cholesterol; TG = triglyceride; UN = urea nitrogen; GLU = glucose; ALP = alkaline phosphatase; ALT = alanine aminotransferase; AST = aspartate aminotransferase. ^2^ G0, diet supplemented with 0% BSFM; G3, diet supplemented with 3% BSFM; G6, diet supplemented with 6% BSFM; G9, diet supplemented with 9% BSFM. Values are mean ± SEM (*n* = 6).

**Table 4 animals-15-00625-t004:** Effects of different black soldier fly larvae meal levels on antioxidant capacity of chickens.

Items ^1^	Groups ^2^			
	G0	G3	G6	G9
21 days of age				
T-AOC (U/mL)	7.53 ± 0.60 ^a^	7.72 ± 0.42 ^ab^	7.97 ± 0.32 ^b^	8.26 ± 0.32 ^b^
GSH-Px (U/mL)	411.49 ± 4.76 ^a^	465.81 ± 8.11 ^b^	470.45 ± 12.68 ^b^	473.15 ± 12.37 ^b^
MDA (nmol/mL)	8.02 ± 0.14 ^Aa^	6.25 ± 0.43 ^b^	5.47 ± 0.45 ^Bb^	5.34 ± 0.28 ^Bb^
T-SOD (U/mL)	138.26 ± 6.28	133.51 ± 4.43	140.56 ± 7.73	141.45 ± 8.20
42 days of age				
T-AOC (U/mL)	7.87 ± 0.52 ^a^	8.28 ± 0.52 ^ab^	8.58 ± 0.39 ^b^	8.52 ± 0.40 ^b^
GSH-Px (U/mL)	414.05 ± 5.28 ^a^	478.53 ± 5.61 ^b^	481.88 ± 5.83 ^b^	478.90 ± 4.64 ^b^
MDA (nmol/mL)	8.11 ± 0.25 ^a^	6.34 ± 0.50 ^b^	6.26 ± 0.14 ^b^	6.10 ± 0.35 ^b^
T-SOD (U/mL)	149.14 ± 4.85	138.61 ± 5.73	149.76 ± 7.54	145.37 ± 6.32

^a,b^ Means that the values within a row with no common letters differ significantly (*p* < 0.05); ^A,B^ Means that the values within a row with no common letters differ significantly (*p* < 0.01). ^1^ T-AOC = total antioxidant capacity; GSHPx = glutathione peroxidase; MDA = malonaldehyde; T-SOD = total superoxide dismutase; CAT = catalase. ^2^ G0, diet supplemented with 0% BSFM; G3, diet supplemented with 3% BSFM; G6, diet supplemented with 6% BSFM; G9, diet supplemented with 9% BSFM. Values are mean ± SEM (*n* = 6).

## Data Availability

The data presented in this study are available upon request from the corresponding author. The data are not publicly available due to company policies.

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
