# Peer review of "Impact of Incorporating Defatted Black Soldier Fly Meal into Diet on Growth Performance, Serum Biochemical Parameters, Nutrient Digestibility, Morphology of the Intestinal Tract, and Immune Index of Brooding Laying Hens"

_animals, 2025, doi:10.3390/ani15050625_

Round 1

Reviewer 1 Report

Comments and Suggestions for Authors

I want to thank authors for their efforts in this work. However, there are some point's needs to be improved.

General comments:

1-      The authors mentioned in the title that BSFM as added in diet of layers, but at the end of the abstract the authors said that BSFM was a replacement for soybean.

(Is the BSFM an addition or replacement?)

Also, as observed in table (1) the BSFM was not a replacement for soybean as the percentage of soybean in treated group was nearly similar and not decreased with the increase of BSFM.

(Please correct this mistake among the manuscript).

2-      All tables need to be improved

a-      The superscript letters need to be in one way among all traits (from the biggest value to the smallest value or the opposite).

In the manuscript some data the superscript is from the big to small values and other from small to big without any key or mention in the materials and methods. This is hard for any reader for manuscript

Please fix this problem

b-          Under each table, please add the key to explain the table (significant and explanation of abbreviations)

Ex: a,b and c means in the same raw having different superscript are significantly different (p≤ 0.05)

Specific comments

Line 19: correct the sentence addition not replacement

Line 43: support the sentence (Plant protein shortages have become increasingly prominent worldwide.) with a reference.

Line 49-51: support the sentence with this reference (Khalifah, A.; Abdalla, S.; Rageb, M.; Maruccio, L.; Ciani, F.; El-Sabrout, K. (2023) Could Insect Products Provide a Safe and Sustainable Feed Alternative for the Poultry Industry? A Comprehensive Review. Animals, 13, 1534.

Line 52-53: this sentence explains the insect requirement for human as food.

 Please add another sentence to explain if insects meet the requirement for poultry.

Line 144: statistical analysis

Please add the mathematical model used in the analysis

Line 220: please remove the sentence (capital letter ……). No need to add this sentence as the table did not have any (p≤ 0.01) significant. (Fix this item among all tables)

Line 247:  this part needs to be revised correctly as the title (Effect of BSFM on nutrient digestibility) was missed before the title of (the intestinal mucosal index). Please fix this part well and correct. Also write the explanation of the abbreviations under each figure.

Author Response

Comments 1: [ The authors mentioned in the title that BSFM as added in diet of layers, but at the end of the abstract the authors said that BSFM was a replacement for soybean.(Is the BSFM an addition or replacement?) Also, as observed in table (1) the BSFM was not a replacement for soybean as the percentage of soybean in treated group was nearly similar and not decreased with the increase of BSFM.(Please correct this mistake among the manuscript).]

Response 1: Agree. Thank you for pointing this out.In this experiment, BSFM was used as an independent protein ingredient to optimize the feed formula for laying hens. This optimization targeted not only soybean meal but also other protein ingredients, with the aim of optimizing the overall feed formula. Therefore, we have modified the last part of the abstract.

Revised version line 46-48: In conclusion, this study indicates that BSFM can serve as a high-quality protein raw material in the process of laying hen breeding, highlighting its potential as a viable insect protein source in the poultry feed industry..

Comments 2: [a-The superscript letters need to be in one way among all traits (from the biggest value to the smallest value or the opposite).In the manuscript some data the superscript is from the big to small values and other from small to big without any key or mention in the materials and methods. This is hard for any reader for manuscript. Please fix this problem

b-Under each table, please add the key to explain the table (significant and explanation of abbreviations) Ex: a,b and c means in the same raw having different superscript are significantly different (p≤ 0.05)]

Response 2: Agree. Thank you for pointing this out. We have made corresponding modifications to all the tables in the article, and added annotations below the tables for the relevant content.

Comments 3: [Line 19: correct the sentence addition not replacement]

Response 3: Agree. Thank you for pointing this out. 

Revised line 30-31: The sentece “while the remaining three groups received diets with soybean oil replaced by 3% (G3 group), 6% (G6 group), and 9% (G9 group) BSFM, respectively.” has been changed to “while the remaining three groups added to 3% (G3 group), 6% (G6 group), and 9% (G9 group) BSFM, respectively.”.

Comments 4: [Line 43: support the sentence (Plant protein shortages have become increasingly prominent worldwide.) with a reference.]

Response 4: Thank you very much for your comments. We have added the corresponding reference in revised line 56.

Comments 5: [Line 49-51: support the sentence with this reference (Khalifah, A.; Abdalla, S.; Rageb, M.; Maruccio, L.; Ciani, F.; El-Sabrout, K. (2023) Could Insect Products Provide a Safe and Sustainable Feed Alternative for the Poultry Industry? A Comprehensive Review. Animals, 13, 1534.]

Response 5: Thank you very much for your comments. We have inserted the reference into the appropriate position.

Comments 6: [Line 52-53: this sentence explains the insect requirement for human as food.Please add another sentence to explain if insects meet the requirement for poultry.]

Response 6: Agree. Thank you for pointing this out. 

Revised line 64-67: The sentence has been changed to “The protein levels and essential amino acid profile of insects meet or are well above the FAO/WHO dietary requirements and it also meets the nutritional requirements for the growth of livestock and poultry [5,9]” 

Comments 7: [Line 144: statistical analysis Please add the mathematical model used in the analysis]

Response 7: Agree. Thank you for pointing this out. 

Revised line 222-228: The sentence has been changed to“one-way ANOVA and Tukey’s multiple comparison of variance using SPSS software (SPSS for Windows, release 19.0, developed by SPSS Inc. Chicago, IL, USA). In order to distinguish statistically significant differences among the means, orthogonal polynomials were used to test linear, quadratic, and cubic responses to dietary levels of BSFM. Finally, the data were presented in the format of mean ± standard deviation (SD), and statistical significance is reported at p < 0.05.” 

Comments 8: [Line 220: please remove the sentence (capital letter ……). No need to add this sentence as the table did not have any (p≤ 0.01) significant. (Fix this item among all tables)]

Response 8: Agree. Thank you for pointing this out.

Revised line 240:“capital letter superscripts were significantly different (P<0.01)”has been deleted and corresponding revisions have been made in accordance with the comments of another reviewer. All the tables have been checked and revised.

Comments 9: [Line 247: this part needs to be revised correctly as the title (Effect of BSFM on nutrient digestibility) was missed before the title of (the intestinal mucosal index). Please fix this part well and correct. Also write the explanation of the abbreviations under each figure.]

Response 9: Agree. Thank you for pointing this out.  

Revised line 282: “As shown in Fig. 1, the apparent digestibility of crude fat in G3 group and G6 group was significantly higher than that in G0 group (p < 0.05). There was no significant difference in apparent digestibility of crude protein, calcium and phosphorus among groups (p > 0.05).” was added. We have also modified the corresponding figures.

Reviewer 2 Report

Comments and Suggestions for Authors

The aim of this study was to investigate the impact of incorporating black soldier fly meal  (BSFM) into the diet of brooding laying hens, with a focus on growth performance, serum biochemical markers, nutrient digestibility, intestinal morphology, and immune responses. The results obtained are important for poultry practice. The research methods used are correct. Sufficient discussion. References selection correct and well used, but they must be prepared in accordance with the instructions for authors.

General comments:

There is no "Simple summary" chapter in the article. Please add it to this article.

In the Materials and methods chapter there is no information about:

  1. Type of building - closed, windowless, with regulated environmental parameters
  2. dimensions of a pen
  3. lighting program (length, intensity, color, type - incandescent, fluorescent)
  4. form of diet (ground?)

Table 1 should be moved to the "Materials and methods" chapter

Explanations for abbreviations used in the tables and descriptions for significance markings should be placed under the tables

Other

For significance, use a low letter "p" in italic (p < 0.05) instead of the "P" in the main article and space

The References section must be done according to the instructions for the authors

Abbreviated name journal is required

Please use a "dot" after each abbreviation, for example "J. Appl. Poult. Res."

Volume numbers are only needed, please remove issue numbers

Please provide page ranges,

For an article with a number, without page numbers, please provide 115917 instead of "p. 115917-„

For page ranges use long dash (-) from the symbol function, instead of short (-) from the keyboard

Detailed comments

L3 on instead of On, with a lower case letter

L4 of instead of Of

L5 and instead of And

L8-11 no initials of the author's name and surname

L12 no author's correspondence phone number

L15 (Hy-Line Brown, commercial flock) instead of current form

L22 feed conversion ratio (FCR) instead of feed-to-gain ratio (F/G)

L32 glutathione peroxidase (GSH-Px) activity and lower malondialdehyde (MDA).. instead of current form

L73 [19,20] instead of [19, 20]

L84 [25] instead of (2023)

L86 Marco et al. [26] instead of current form, delete (2015)

L87 delete (2017)

L120 (Hy-Line Brown, commercial flock) instead of HY-LINE variety brown

L123 Hy-Line Brown

L139 33 °C, space after 33

L141 21 °C, space after 21

L207 FCR instead of F/G

In Table 2 FCR instead of F/G

L234 birds instead of chicks, the term chicks is usually used until 14 days of age

L236 T-AOC instead of TAC

L240 with group G3 there is also a significant difference

L247 1&2, "1" instead of "!"

L251 IL6, group G9 has a significant difference designation

In Figures 1 and 2, please remove the “NS” designations – reduced clarity

IN Table 4, superscript for “B” for group G6 – GSH-Px trait

L281 delete (2024)

L296 [33,34] no space

L300 [35,36]

L304 [37,38]

L317 [42,43]

L349 Secci et al. dot after “al.”

L364 Cutrignelli et al. [….] – no reference number

L385 space before BSF

L387 Schiavone et al. dot after “al.”

L400 [54,55]

L418 “FCR of birds” instead of F/G of chicks

L449 and others fill in missing page ranges, remove issue number, etc. see General comments and instructions for authors

Author Response

Comments 1: [There is no "Simple summary" chapter in the article. Please add it to this article.]

Response 1: Thank you for pointing this out. The "Simple summary" part has been completed in the revised version.

Revised version line 14-24: Simple summary:This study explored the use of defatted black soldier fly larvae meal (BSFM) as a sustainable protein alternative to soybean meal in the diets of brooding laying hens. Over 42 days, 480 chicks were divided into four groups fed diets with 0%, 3%, 6%, or 9% BSFM. Results showed that hens fed 3% BSFM had higher daily weight gain and improved feed efficiency during early growth phases. BSFM supplementation also enhanced blood protein levels, antioxidant capacity, and immune function while reducing fat and nitrogen waste in the blood. Nutrient digestion and gut health remained stable, with no negative impacts on feed intake or bone development. These findings suggest that BSFM can effectively replace soybean meal in poultry feed, offering a sustainable protein source that supports animal growth, health, and environmental goals. This research provides valuable insights for reducing reliance on traditional protein sources in the poultry industry.

Comments 2: [In the Materials and methods chapter there is no information about:

Type of building - closed, windowless, with regulated environmental parameters

dimensions of a pen

lighting program (length, intensity, color, type - incandescent, fluorescent)

form of diet (ground?).]

Response 2: Agree. Thank you for pointing this out. We have supplemented the laying hen breeding conditions.

Revised version line 163-166:The sentences “An enclosed chicken coop was adopted, with chickens raised in cages and artificial ventilation employed. In the first three days, 24 - hour lighting was provided, and afterwards, the lighting duration was maintained at 10 hours per day. White light tubes were used for illumination.” were added.

Comments 3: [Table 1 should be moved to the "Materials and methods" chapter: Explanations for abbreviations used in the tables and descriptions for significance markings should be placed under the tables]

Response 3: Thank you for your comments. Table 1 has been moved to the "Materials and methods" chapter.

Table 1. Composition and nutrient levels of experimental diets(as fed basis).

Items (%)

inclusion levels1, %

0

3

6

9

Ingredients

Corn

65.01

67.66

67.55

67.21

Soybean meal

26.37

22.30

22.63

20.07

Corn gluten meal

3.00

3.00

0.44

0.00

Defatted BSFM

0.00

3.00

6.00

9.00

Soybean oil

1.35

0.00

0.00

0.00

Limestone

1.40

1.09

0.54

0.83

Dicalcium phosphate

1.50

1.58

1.64

1.76

L-Lysine·HCl

0.27

0.28

0.10

0.04

DL-Methionine

0.10

0.09

0.10

0.09

Vitamin and mineral

premix

1.00

1.00

1.00

1.00

Total

100.0

100.0

100.0

100.0

Nutrient levels*

Metabolizable energy,

MJ/kg

12.13

12.13

12.13

12.13

Crude protein, %

19.00

19.00

19.00

19.00

Calcium, %

0.87

0.98

1.00

1.30

Non-phytate phosphorus, %

0.41

0.42

0.42

0.43

Lysine, %

1.05

1.05

1.05

1.05

Methionine, %

0.41

0.41

0.41

0.41

* The crude protein, lysine and methionine levels were measured value; other nutrient levels were calculated values.2G0, diet supplemented with 0% BSFM; G3, diet supplemented with 3% BSFM; G6, diet supplemented with 6% BSFM; G9, diet supplemented with 9% BSFM.

Comments 4: [For significance, use a low letter "p" in italic (p < 0.05) instead of the "P" in the main article and space]

Response 4: Agree. Thank you for pointing this out. We have changed the capital "P" to lowercase italic "p" throughout the manuscript.

Comments 5: [The References section must be done according to the instructions for the authors

Abbreviated name journal is required

Please use a "dot" after each abbreviation, for example "J. Appl. Poult. Res."

Volume numbers are only needed, please remove issue numbers

Please provide page ranges,

For an article with a number, without page numbers, please provide 115917 instead of "p. 115917-„

For page ranges use long dash (-) from the symbol function, instead of short (-) from the keyboard.]

Response 5: Agree. Thank you for pointing this out. We have revised the reference section in accordance with the requirements specified by the author.

Comments 6: [L3 on instead of On, with a lower case letter]

Response 6: Agree. Thank you for pointing this out. 

Revised line 3: “On” has been replaced with “on”.

Comments 7: [L4 of instead of Of?]

Response 7: Thank you for your comments. 

Revised line 4: “Of” has been replaced with “of”.

Comments 8: [L5 and instead of And.]

Response 8: Agree. Thank you for pointing this out. 

Revised line 5: “And” has been replaced with “and”.

Comments 9: [L8-11 no initials of the author's name and surname.]

Response 9: Agree. Thank you for pointing this out. We have made revisions in accordance with your suggestions.

 Comments 10: [L12 no author's correspondence phone number]

Response 10: Agree. Thank you for pointing this out. We have made revisions in accordance with your suggestions.

Comments 11: [L15 (Hy-Line Brown, commercial flock) instead of current form.]

Response 11: Agree. Thank you for pointing this out. We have made revisions in accordance with your suggestions.

Comments 12: [L22 feed conversion ratio (FCR) instead of feed-to-gain ratio (F/G).]

Response 12: Agree. Thank you for pointing this out. We have made revisions in accordance with your suggestions.

Comments 13: [L32 glutathione peroxidase (GSH-Px) activity and lower malondialdehyde (MDA).. instead of current form]

Response 13: Agree. Thank you for pointing this out. We have made revisions in accordance with your suggestions.

Revised line 47: The sentence has been changed to “glutathione peroxidase (GSH-Px) activity and lower malondialdehyde (MDA)”.

Comments 14: [L73 [19,20] instead of [19, 20].]

Response 14: Agree. Thank you for pointing this out.We have made revisions in accordance with your suggestions.

Comments 15: [L84 [25] instead of (2023).]

Response 15: Agree. Thank you for pointing this out. We have made revisions in accordance with your suggestions. “(2023)” has been deleted.

Comments 16: [L86 Marco et al. [26] instead of current form, delete (2015)]

Response 16: Agree. Thank you for pointing this out. We have made revisions in accordance with your suggestions. “(2015)” has been deleted.

Comments 17: [L87 delete (2017)]

Response 17: Agree. Thank you for pointing this out. We have made revisions in accordance with your suggestions. “(2017)” has been deleted.

Comments 18: [L120 (Hy-Line Brown, commercial flock) instead of HY-LINE variety brown.]

Response 18: Agree. Thank you for pointing this out. We have made revisions in accordance with your suggestions. “HY-LINE variety brown” has been replaced with “Hy-Line Brown, commercial flock”.

Comments 19: [L123 Hy-Line Brown.]

Response 19: Agree. Thank you for pointing this out. We have made revisions in accordance with your suggestions.“Hyline brown” has been replaced with “Hy-Line Brown”.

Comments 20: [L139 33 °C, space after 33]

Response 20: Agree. Thank you for pointing this out. We have made revisions in accordance with your suggestions.“33℃” has been replaced with “33 â„ƒ”.

Comments 21: [L141 21 °C, space after 21.]

Response 21: Agree. Thank you for pointing this out. We have made revisions in accordance with your suggestions.“21℃” has been replaced with “21 ℃”.

Comments 22: [L207 FCR instead of F/G.]

Response 22: Agree. Thank you for pointing this out. We have made revisions in accordance with your suggestions.“F/C” has been replaced with “FCR”.

Comments 23: [In Table 2 FCR instead of F/G.]

Response 23: Agree. Thank you for pointing this out. We have made revisions in accordance with your suggestions.“F/C” has been replaced with “FCR”.

Comments 24: [L234 birds instead of chicks, the term chicks is usually used until 14 days of age.]

Response 24: Agree. Thank you for pointing this out. We have made revisions in accordance with your suggestions.“chicks” has been replaced with “birds”.

Comments 25: [L236 T-AOC instead of TAC.]

Response 25:Agree. Thank you for pointing this out. We have made revisions in accordance with your suggestions.“TAC” has been replaced with “T-AOC”.

Comments 26: [L240 with group G3 there is also a significant difference.]

Response 26:Agree. Thank you for pointing this out. We have made revisions in accordance with your suggestions.“G3” has been added.

Comments 27: [L247 1&2, "1" instead of "!".]

Response 27:Agree. Thank you for pointing this out. We have made revisions in accordance with your suggestions.“!” has been replaced with “1”.

Comments 28: [L251 IL6, group G9 has a significant difference designation.]

Response 28: Thank you for pointing this out. We have made revisions in accordance with your suggestions. The sentence has been changed to “G9 group has a significant difference designation in the IL-6 levels compared with G0 (p < 0.01), the IL-6 and TNF-α levels were found to be comparable across other groups (p > 0.05). 

Comments 29: [In Figures 1 and 2, please remove the “NS” designations – reduced clarity.]

Response 29:Thank you for pointing this out. Based on the suggestions from you and other reviewers, we have modified Figure 1 and Figure 2 to enhance their clarity. At the same time, these figures now meet the relevant graphical specifications.

Comments 30: [IN Table 4, superscript for “B” for group G6 – GSH-Px trait.]

Response 30:Agree. Thank you for pointing this out. We have made revisions in accordance with your suggestions.

Comments 31: [L281 delete (2024).]

Response 31:Agree. Thank you for pointing this out. We have made revisions in accordance with your suggestions.

Comments 32: [L296 [33,34] L300 [35,36] L304 [37,38] L317 [42,43]no space.]

Response 32: Agree. Thank you for pointing this out. We have made revisions in accordance with your suggestions.

Comments 33: [L349 Secci et al. dot after “al.”.]

Response 33: Agree. Thank you for pointing this out. We have made revisions in accordance with your suggestions.

Comments 34: [L364 Cutrignelli et al. [….] – no reference number.]

Response 34: Agree. Thank you for pointing this out. We have made revisions in accordance with your suggestions.

Comments 35: [L385 space before BSF.]

Response 35: Agree. Thank you for pointing this out. We have made revisions in accordance with your suggestions.

Comments 36: [L387 Schiavone et al. dot after “al.”.]

Response 36: Agree. Thank you for pointing this out. We have made revisions in accordance with your suggestions.

Comments 37: [L400 [54,55].]

Response 37: Agree. Thank you for pointing this out. We have made revisions in accordance with your suggestions.

Comments 38: [L418 “FCR of birds” instead of F/G of chicks.]

Response 38: Agree. Thank you for pointing this out. We have made revisions in accordance with your suggestions.

Comments 39: [L449 and others fill in missing page ranges, remove issue number, etc. see General comments and instructions for authors.]

Response 39: Agree. Thank you for pointing this out. We have made revisions in accordance with your suggestions.

Round 2

Reviewer 1 Report

Comments and Suggestions for Authors

I want to thank the authors for their response to the reviewer comments.

Author Response

Thank you for your approval. We have made revisions in accordance with the comments of other reviewers. Meanwhile, we have carefully reviewed the entire manuscript and made modifications in the corresponding positions.

Reviewer 2 Report

Comments and Suggestions for Authors

The aim of this study was to investigate the impact of incorporating black soldier fly meal  (BSFM) into the diet of brooding laying hens, with a focus on growth performance, serum biochemical markers, nutrient digestibility, intestinal morphology, and immune responses. The results obtained are important for poultry practice. The research methods used are correct. Sufficient discussion. References selection correct and well used, but they must be prepared in accordance with the instructions for authors.

General comments:

In the Materials and methods chapter there is no information about:

  1. dimensions of a cage (L161)

Detailed comments

L20 birds instead of hens

L79 [12,13] without spaces

L93 Khan et al. [26]

L97 delete [26]

L101 De Marco instead of D Marco

In Table 1 Premix with a capital letter

In References section

full author's composition is required for each item instead of "et al."

Please use a "dot" after each abbreviation, for example "J. Appl. Poult. Res." for each item

For page ranges use long dash (-) from the symbol function, instead of short (-) from the keyboard

Author Response

Comments 1: [In the Materials and methods chapter there is no information about: dimensions of a cage (L161)]

Response 1: Thank you for pointing this out. 

Revised version line 161: The sentence has changed to “An enclosed chicken coop was adopted, each replicate uses one chicken cage (length×width×height: 87×45×24 cm) and artificial ventilation was employed.”. It specifies the dimensions of the chicken cage used.

Comments 2: [L20 birds instead of hens]

Response 2: Agree. Thank you for pointing this out. 

Revised version line 20: “hens” has been replaced with “birds”.

Comments 3: [L79 [12,13] without spaces]

Response 3: Thank you for your comments. We have deleted the space.

Comments 4: [L93 Khan et al. [26]]

Response 4: Agree. Thank you for pointing this out. “Khan et al.” has been replaced with “Khan et al. [26]”.

Comments 5: [L97 delete [26]]

Response 5: Agree. Thank you for pointing this out. We have deleted the “[26]”.

Comments 6: [L101 De Marco instead of D Marco]

Response 6: Agree. Thank you for pointing this out. 

“D Marco” has been replaced with “De Marco”.

Comments 7: [In References section

full author's composition is required for each item instead of "et al."

Please use a "dot" after each abbreviation, for example "J. Appl. Poult. Res." for each item

For page ranges use long dash (-) from the symbol function, instead of short (-) from the keyboard]

Response 7: Thank you for your comments. 

We have made revisions in accordance with your suggestions.
